# Differences in Acute Expression of Matrix Metalloproteinases-9, 3, and 2 Related to the Duration of Brain Ischemia and Tissue Plasminogen Activator Treatment in Experimental Stroke

**DOI:** 10.3390/ijms25179442

**Published:** 2024-08-30

**Authors:** Dong Wang, Sofiyan Saleem, Ryan D. Sullivan, Tieqiang Zhao, Guy L. Reed

**Affiliations:** Department of Medicine, University of Arizona College of Medicine, Phoenix, AZ 85004, USA; dwang16@arizona.edu (D.W.); ssaleem1@arizona.edu (S.S.); ryansullivan@arizona.edu (R.D.S.); tqzhao@yahoo.com (T.Z.)

**Keywords:** ischemic stroke, recombinant tissue plasminogen activator, middle cerebral artery occlusion, metalloproteinases

## Abstract

Matrix metalloproteinases (MMPs) such as MMP-9, 3, and 2 degrade the cellular matrix and are believed to play a crucial role in ischemic stroke. We examined how the duration of ischemia (up to 4 h) and treatment with recombinant tissue plasminogen activator altered the comparative expression of these MMPs in experimental ischemic stroke with reperfusion. Both prolonged ischemia and r-tPA treatment markedly increased MMP-9 expression in the ischemic hemisphere (all *p* < 0.0001). The duration of ischemia and r-tPA treatment also significantly increased MMP-2 expression (*p* < 0.01–0.001) in the ischemic hemisphere (*p* < 0.01) but to a lesser degree than MMP-9. In contrast, MMP-3 expression significantly decreased in the ischemic hemisphere (*p* < 0.001) with increasing duration of ischemia and r-tPA treatment (*p* < 0.05–0001). MMP-9 expression was prominent in the vascular compartment and leukocytes. MMP-2 expression was evident in the vascular compartment and MMP-3 in NeuN+ neurons. Prolonging the duration of ischemia (up to 4 h) before reperfusion increased brain hemorrhage, infarction, swelling, and neurologic disability in both saline-treated (control) and r-tPA-treated mice. MMP-9 and MMP-2 expression were significantly positively correlated with, and MMP-3 was significantly negatively correlated with, infarct volume, swelling, and brain hemorrhage. We conclude that in experimental ischemic stroke with reperfusion, the duration of ischemia and r-tPA treatment significantly altered MMP-9, 3, and 2 expression, ischemic brain injury, and neurological disability. Each MMP showed unique patterns of expression that are strongly correlated with the severity of brain infarction, swelling, and hemorrhage. In summary, in experimental ischemic stroke in male mice with reperfusion, the duration of ischemia, and r-tPA treatment significantly altered the immunofluorescent expression of MMP-9, 3, and 2, ischemic brain injury, and neurological disability. In this model, each MMP showed unique patterns of expression that were strongly correlated with the severity of brain infarction, swelling, and hemorrhage.

## 1. Introduction

Matrix metalloproteinases (MMPs), a group of proteolytic zinc-dependent enzymes, are important for normal brain function [1,2]. MMPs have been investigated for their role in many cerebral pathologies, such as stroke and the disruption of the blood–brain barrier (BBB) post stroke [3]. Among the 23 MMPs that have been identified to date, MMP-2 and MMP-9 are the most widely studied in stroke due to their critical role involved in pathogenesis of BBB breakdown and subsequent vasogenic edema [4]. MMP-2 and MMP-9 are both gelatinases that are capable of processing proteins in the extracellular matrix (ECM) as well as receptors, cytokines, and growth factors that are involved in migration, inflammation, angiogenesis, and other processes [5]. MMP-9 causes degradation of ZO-1, a tight junction protein maintaining the integrity of BBB, weakens the plaque fibrous cap, and plays a crucial role in the proteolytic degradation of the BBB [6]. MMP-3 is a member of the stromelysin subgroup of MMPs and has been found to degrade many ECM proteins [7]. In addition, MMP-3 can activate MMP-9 following cerebral ischemia [8].

The expression of MMPs under physiological conditions is generally very low; the main constitutive enzymes in the brain are MMP-2 and membrane-type MMP (MMP-14), which present normally in astrocytes associated with blood vessels [9]. In contrast, MMP-3 and MMP-9 are inducible [10]. The MMPs are tightly regulated at both the transcriptional and post-transcriptional levels, and they respond to ischemic brain injury differently [4]. How the degree of ischemic injury/ischemic time affects MMP levels has not been well defined, which is necessary for translating these observations into clinical utility.

The detrimental roles of matrix metalloproteinases (MMPs), particularly MMP-9 and MMP-2, have been well-documented in both ischemic and hemorrhagic strokes in humans [11,12]. These enzymes are overexpressed at various time points post-stroke, ranging from hours to days [12,13]. Their expression has been linked to stroke severity, infarct size, and clinical outcomes [14]. However, due to the variability in endpoints and targeted MMPs among clinical studies, arising from differences in clinical settings, it remains difficult to construct a comprehensive timeline of MMP-9, MMP-3, and MMP-2 expression. Moreover, the combined impact of tissue plasminogen activator (t-PA) treatment and ischemic injury remains underexplored. One translational study demonstrated that t-PA treatment selectively induced MMP-3 expression in endothelial cells within the ischemic-damaged area in a mouse stroke model [14]. This finding highlights the overlapping effects in the natural progression of stroke. Despite these valuable insights, assembling a complete picture is challenging due to variations in patient populations, clinical assessment timelines post-stroke, and evaluated parameters.

Recombinant tissue plasminogen activator (r-tPA) remains the only approved therapy for stroke; however, only 5–10% of stroke patients are treated with r-tPA because of restricted inclusion criteria [15]. Thrombolysis with r-tPA after acute ischemic stroke has been limited by the short-recommended time window of <4.5 h of symptoms [16,17]. Within the r-tPA treatment window, r-tPA treatment within 1.5 h of symptom onset has been shown to have the best outcome, followed by treatment within 1.5 to 3.0 h, and then treatment within 3.0 to 4.5 h [18]. Beyond this therapeutic window, r-tPA appears to lose its beneficial properties and increases the risk of brain edema and hemorrhagic transformation (HT) by damaging the basal lamina of the blood vessels and disrupting the BBB [19]. It activates MMPs, such as MMP-9, which was shown to be elevated in venous blood from stroke patients that received tPA treatment [4]. MMP-3 also contributes to HT in hyperglycemic conditions [20]. However, the influence of the duration of ischemia on r-tPA treatment-induced MMPs expression post-stroke remains unclear.

The focus of the present study was to determine the time-dependent effects of the duration of ischemia and r-tPA treatment on brain injury, neurobehavioral outcomes, and the expression of MMPs (MMP-9, MMP-2, and MMP-3) in cerebral ischemia with reperfusion. By comparing these changes simultaneously, we gained a more precise view of how the duration of ischemia and r-tPA treatment affects MMPs expression and neurologic outcomes post stroke.

## 2. Results

### 2.1. Effect of Ischemia Time and r-tPA Treatment on MMPs Expression in the Brain

We examined the effect of different durations of ischemia (1, 2, and 4 h) and treatment with saline or r-tPA on MMP expression. Occlusion of the middle cerebral artery was confirmed by ≥80% drop in cerebral blood flow from baseline as measured by laser Doppler flowmetry in both saline control and r-tPA-treated groups. The effects of ischemic time and therapy on the expression of MMP-9, -3, and -2 were analyzed 24 h after onset of ischemia. MMP-9 expression increased with the duration of ischemia time in the control (ctl) or r-tPA-treated mice (1 h vs. 2 or 4 h, *p* < 0.01 in ctl, *p* < 0.0001 in r-tPA, Figure 1). When compared with ctl, r-tPA-treated mice had significantly more MMP-9 expression in the ischemic hemisphere at 2 and 4 h of ischemia (Figure 1a,d, *p* < 0.001 and *p* < 0.0001). The expression of MMP-9 in non-stroke hemisphere showed a minimal, non-significant increase in ctl group regardless of ischemic time, while r-tPA-treated mice showed a mild increase at 4 h.

MMP-3 immunostaining was evident throughout both hemispheres but appeared to decline in the ischemic hemisphere by comparison to the contralateral hemisphere (Figure 1b,e). Treatment with r-tPA significantly decreased MMP-3 expression of the ischemic hemisphere in the ischemic 2 h group by comparison to the control group (Figure 1b,e, *p* < 0.001), but this difference was no longer significant when the ischemic time was extended to 4 h.

There was a significant increase in MMP-2 expression in the stroke hemisphere of the r-tPA-treated mice (2 h of ischemia, but not 1 or 4 h) when compared with the control mice (Figure 1c,f, *p* < 0.0001). In contrast, MMP-2 expression in the non-stroke hemisphere was not significantly changed when comparing control and r-tPA-treated mice at all three ischemic timepoints (*p* > 0.05 at 1, 2, and 4 h).

A quantitative overall analysis (Table 1) shows that MMP-9 expression was significantly increased by ischemic time (*p* < 0.0001), r-tPA treatment (*p* < 0.0001), and hemisphere location (ischemic vs. non-ischemic; *p* < 0.0001). A similar, but less pronounced pattern, was seen in the MMP-2 expression, which was increased in the ischemic hemisphere (*p* = 0.007), by ischemic time (*p* = 0.0003), and by r-tPA treatment (*p* = 0.0016). MMP-3 expression was also altered by ischemic time, treatment, and ischemic vs. non-ischemic hemisphere (Table 1, *p* < 0.0001, *p* = 0.015, and *p* < 0.001, respectively). For MMP-9, the duration of ischemia and the brain hemisphere (ischemic vs. non-ischemic) contributed most to the total variation in expression observed in these studies, followed by r-tPA treatment. For MMP-3, ischemic time and brain hemisphere were responsible for the greatest amount of total variation in expression, and treatment accounted for <5%. For MMP-2, the duration of ischemic time accounted for most of the total variation, while the brain hemisphere and treatment contributed <10% each (see Appendix A).

### 2.2. Cellular Specificity of MMPs Expression in the Brain

To identify the sources of MMPs expression, we used various cellular markers. MMP-9 expression was predominantly found in the ischemic core and appeared to be localized within blood vessels that were immunostained for Type IV collagen (Figure 2a, yellow arrowhead), suggesting a vascular compartment location. Interestingly, MMP-9 was also co-stained with neutrophil and macrophage cell markers, Ly6G and CD68 (Figure 2b,c), consistent with inflammatory cell expression. MMP-2 was only detected in the periphery of blood vessels that were immuno-stained for Type IV collagen (Figure 2d). MMP-3 did not have a vascular pattern but was co-stained with NeuN (Figure 2e), a neuronal marker, suggesting a neuronal cell source of MMP-3. In addition, MMP-3 showed a cytoplasmic location on neurons (Figure 2e, yellow arrowhead).

### 2.3. Correlation between MMPs Expression and Ischemic Stroke Outcomes

To define the potential relationship between the expression levels of MMPs and various outcomes, we examined correlations using either Pearson or Spearman tests (Figure 3). MMP-9 expression levels in both the control and r-tPA groups were positively correlated with infarct volume (Figure 3a) while only r-tPA group’s MMP-9 levels were positively correlated with swelling and hemorrhage (Figure 3d,g). MMP-2 levels showed a similar trend to MMP-9. MMP-2 expression levels in both the control and r-tPA groups were correlated with infarct volume (Figure 3c). However, in contrast to MMP-9, only the control group’s MMP-2 levels were positively correlated with swelling, and there was no significant relationship between hemorrhage and MMP-2 levels in either the control or r-tPA group (Figure 3f,i). MMP-3 was different from MMP-9 and MMP-2. MMP-3 expression levels in both the control and r-tPA groups were negatively correlated with infarct volume and swelling (Figure 3b,e) while only the control group’s MMP-3 levels were negatively correlated with hemorrhage (Figure 3h).

### 2.4. Time-Related Effects of Ischemia and r-tPA Treatment on Neurobehavioral and Histological Outcomes Post Stroke

Twenty-four hours after ischemia, neurobehavioral tests were performed, and brains were examined. The effect of ischemia duration and time are shown graphically in Figure 4. When analyzed by two-way ANOVA, the duration of ischemia significantly increased infarction (Figure 4a, *p* < 0.001), swelling (Figure 4b, *p* < 0.001), and brain hemorrhage (Figure 4c, *p* < 0.001). The duration of ischemia also adversely affected neurobehavioral outcome as indicated by worsened scores on the neurobehavioral and corner tests (Figure 4d,e, *p* < 0.0001). When compared to the control (saline treatment), r-tPA treatment significantly increased brain infarction (*p* < 0.05) and swelling (*p* < 0.05).

## 3. Discussion

We examined the effect of ischemia and r-tPA treatment at different ischemic time points (1, 2, and 4 h) on the comparative expression levels of MMP-9, -3, and -2 in the brain post stroke. Ischemic time is considered as one of the most important factors that directly influences clinical outcomes post stroke [17]. Consistent with our previous study [21], we found here that both ischemia and r-tPA enhanced brain injury and neurologic deficits after experimental ischemic stroke, especially at 2 and 4 h after r-tPA treatment, compared with saline control treatment. Additionally, r-tPA worsened ischemic injury as reflected by increased brain infarction and swelling. The expression of MMP-9 and MMP-2 was upregulated, while MMP-3 was downregulated with increasing duration of ischemia and r-tPA treatment. MMP-9 was consistently increased with the duration of ischemia. MMP-9 expression was attributed to blood cells, Ly6G+ neutrophils, and CD68+ monocytes, while MMP-2 was only identified in blood vessel compartment and MMP-3 was expressed in NeuN+ neurons.

No previous reports have clearly compared the time-dependent expression of different MMPs (MMP-9, MMP-3, MMP-2) after r-tPA treatment in ischemic stroke and their presence in neuronal or vascular compartments [3]. MMPs, in particular MMP-9, 2, and 3, may play an important role in tPA-associated hemorrhagic complications [3]. However, there is controversy regarding the role of MMP-2 and MMP-3 after r-tPA treatment in ischemic stroke [20,22,23,24]. We found that the expression of MMP-9 in the stroke hemisphere of r-tPA-treated mice was highly dependent on ischemic time and was significantly greater in r-tPA-treated mice than control mice with 2 h and 4 h ischemia. Similarly, MMP-2 significantly increased with the duration of expression in the ischemic hemisphere after r-tPA treatment. Both MMP-9 and MMP-2 played a key role in the disruption of the BBB with subsequent hemorrhage, brain infarction, and edema after cerebral ischemia [4]. Such alterations in MMP-9 and MMP-2 expression are consistent with the increased infarct volume, hemorrhage, and swelling. Further, both MMP-2 and MMP-9 may attack the components of the basal lamina around the cerebral blood vessels, such as type IV collagen, laminin, and fibronectin, which contribute to the hemorrhagic transformation during the early stage of cerebral ischemia and reperfusion [25]. Immunostaining showed that MMP-9 and MMP-2 were expressed in vascular compartment, suggesting they may play an important role in tPA treatment-associated enhancement of infarct volume, hemorrhage, and swelling. Following disruption of the BBB, peripheral immune cells are reported to migrate into the brain to enhance the neuroinflammatory response, and to hasten the development of vasogenic edema [26]. Consistent with other findings that infiltrated neutrophils expressed MMP-9 post stroke [27,28], our double-immunostaining also showed that MMP-9 was found in infiltrated neutrophil and monocytes outside the vascular compartment, whereas MMP-2 was only detected in the vascular compartment. MMP-2 has been considered as the “first responder” MMP post-stroke [18], and MMP-2 caused reversible BBB disruption [2]. In contrast, elevated MMP-9 caused complete degradation of the basal lamina [29] and tight junction components, which eventually lead to gross barrier disruption [6]. This suggests that it is reasonable to expect a more severe secondary cascade in blood vessels with high MMP-9 expression following stroke and r-tPA treatment. Thus, among the gelatinases, MMP-9 may play a greater role in BBB disruption and poor neuronal outcomes after cerebral ischemia and r-tPA treatment. Recent clinical studies have linked MMP-9 expression in the blood with infarct growth and hemorrhagic transformation after ischemic stroke [30].

In contrast to MMP-9 and MMP-2 expression after r-tPA treatment in ischemic stroke, MMP-3 levels dropped post stroke without r-tPA treatment and further decreased following r-tPA treatment. Suzuki et al. found that MMP-3 may play an important role in ICH induced by tPA treatment of ischemic stroke in mice [31]. In addition, in a mouse stroke model, tPA treatment selectively induced MMP-3 expression in injured endothelial cells [3]. However, we found that MMP-3 was mainly expressed in NeuN+ cells but not endothelial cells, neutrophils, or monocytes. This suggests that MMP-3 might not be involved in the degradation of the blood vessel barrier but related to neuronal cell death post stroke. A prior study showed that MMP-3 may have neuroprotective functions by cleaving the transmembrane form of the Fas ligand (Fas-L) to block the apoptotic pathway [32]. Considerable evidence showed that the Fas, FasL, and TNF-mediated extrinsic apoptotic signaling cascade plays an important role in neuronal cell death after cerebral ischemia [33]. In addition, demyelination is a major component of white matter injury and contributes significantly to neurobehavioral and cognitive impairments following stroke [34]. MMP-3 also had beneficial effects on remyelination of pyramidal neurons in the cortex and hippocampus [35]. We find that tPA treatment further decreases MMP-3 levels following cerebral ischemia, which may explain the bigger infarct volume and worsen neurologic dysfunction in tPA group. Furthermore, the release of MMP-3 from apoptotic neurons may trigger microglial activation and inflammatory reactions as well as exacerbation of neuronal apoptosis, which leads to rapid phagocytosis of apoptotic neurons [36]. Such positive feedback could further extend brain injury after r-tPA treatment. Thus, future studies need to investigate the role of MMP-3 in neuron survival post ischemic stroke.

With the advent of penumbral imaging and thrombectomy, patients are being treated with r-tPA and reperfusion therapy after many hours of ischemia [30]. In these patients, the breakdown of the blood–brain barrier, hemorrhage, and infarct growth continue to be a challenge, which is related to ischemic time and MMP expression [30]. Our data show (Table 1) that prolonged ischemia and r-tPA therapy separately affect the expression of MMP-2,3, and 9 in the ischemic hemisphere. In addition, ischemic time and r-tPA interact to alter MMP expression (Appendix A). Therefore, when administering r-tPA to stroke patients, careful consideration should be given to the duration of ischemia. For patients who receive delayed tPA treatment, combining an MMP inhibitor may help reduce the risk of BBB leakage and hemorrhagic transformation. Success with similar approaches strengthens the potential for this strategy. For example, combining minocycline with tPA extended the therapeutic window from 3 h to 6 h in an embolic ischemia stroke rat model [37]. Similarly, combining atorvastatin with tPA reduced MMP-9 levels, lowered the incidence of hemorrhagic transformation, and extended the therapeutic window to 6 h in a rat stroke model [38]. These findings suggest that combining MMP inhibitors with tPA may be a promising strategy to further expand the therapeutic window for stroke patients.

The application of MMP inhibitors for stroke treatment remains premature due to the complexity of MMP expression after stroke and because most available MMP inhibitors are a broad spectrum rather than being limited to specific MMPs. At the same time, there is good evidence of MMP-9 upregulation in human brain tissues and in the blood after stroke, and increased blood levels are linked to late hemorrhagic transformation [14]. MMP-9 has also been linked with inflammatory responses in various diseases, including myocardial infarction, stroke, Alzheimer’s disease, multiple sclerosis, and tumors. As with ischemic stroke, the application of specific MMP-9 inhibitors to these conditions will require greater knowledge of time-dependent expression and function during the disease process.

### Limitation

The pathogenesis of ischemic stroke is a complex, dynamic, and multifactorial process. There are important limitations to our study. Although individual MMP expression had been examined at the prolonged time points, such as 24 h, 5, 7, and even 14 days, post stroke [39], MMP-9, MMP-3, and MMP-2 had not been compared simultaneously and we chose the 24 h time point to detect acute changes. Second, the transient middle cerebral artery occlusion (MCAO) model was used in the present study, which allowed us to examine the effect of the duration of ischemia at specific time points. Other stoke models, such as the thromboembolic stroke model [40], more closely mimic human stroke but do not allow for an analysis of specific effects of ischemic duration [41]. Third, fibrinolysis is associated with r-tPA’s favorable effect and is only effective within a short time window in mice (~15–30 min), which was not examined in these studies [42]. Fourth, the present study used immunohistology to comparatively measure the expression level of MMP proteins and their cellular source. We and others have assessed MMP enzymatic activity post-stroke, and extensive transcript analyses have also been performed [43,44,45]. Fifth, future studies may be needed to examine the comparative expression of MMPs -8, -10, and -13 and other MMPs that may also play a role in general ischemic brain injury [46]. Sixth, while sex differences are increasingly recognized in ischemic stroke [47], we are unable to assess the effect of sex on MMP expression as our current study was restricted to male mice. Finally, our studies have correlated MMP expression with neurologic outcomes; when specific inhibitors are available, it will be possible to test the causative relationship between these molecules and stroke outcomes.

## 4. Materials and Methods

### 4.1. Animals

All animal procedures were performed in compliance with institutional guidelines of the Animal Ethics Committee of the University of Arizona, which approved these experiments. A total of 36 male C57BL/6J (Jackson laboratory, Bar Harbor, ME, USA), ages 8–12 weeks, were randomized to experimental groups. Animals were housed in compliance with standard conditions (12 h-day–night cycle, 60% humidity, 22 °C room temperature) and had free access to food pellets and water. Carbon dioxide euthanasia was performed at the study endpoints, followed by blood collection via cardiocentesis and saline perfusion, which were performed as previously described [48].

### 4.2. Transient Middle Cerebral Artery Occlusion and Experimental Groups

Adult wild type (WT) male C57BL/6J mice (~23–28 g) were anesthetized with isoflurane. Experiments were performed by a blinded investigator. Transient focal cerebral ischemia was induced by middle cerebral artery occlusion (MCAO) with an intraluminal filament as described previously [49,50]. Each animal was maintained with continuous-flow 1.5–2.0% isoflurane (after induction with 4.0% isoflurane) in oxygen-enriched air via a nose cone. The core body temperature (rectal) was maintained at 37.0 ± 0.5 °C by a heating pad. Relative cerebral blood flow (CBF) was measured by laser-Doppler flowmetry (LDF, AD Instruments, Oxford Optonix, Banbury, UK) with a flexible fiberoptic probe affixed to the skull over the parietal cortex perfused by the MCA (2 mm posterior and 6 mm lateral to the bregma). Under aseptic conditions, the neck and carotid bifurcation were dissected, and the common carotid artery was temporarily ligated. A silicon-coated 7–0 Ethilon nylon monofilament was purchased (Doccol corp., Sharon, MA, USA) and inserted to occlude the MCA. The filament was advanced through an incision in the external carotid artery stump through the internal carotid artery to the origin of the MCA; successful occlusion was documented by a decrease in the laser-Doppler signal of at least 80%. In these experiments, the filament was left in position for 1–4 h, and the mice were recovered for 24 h. During occlusion, the neck wound was closed with sutures, anesthesia was discontinued, and the animals were transferred to a temperature-controlled cage to maintain the body temperature at 37.0 ± 0.5 °C. At the time of reperfusion, the mouse was briefly re-anesthetized with isoflurane, and reperfusion was achieved by slowly withdrawing the filament. A dose of analgesic (buprenorphine SR:1 mg/mL) was given subcutaneously prior to surgery and as needed post-surgery to prevent pain. The intravascular volume was supported by giving 0.5 mL of 37 °C saline intraperitoneally post-surgery and every day for the duration of the study. Mice were housed in cages with unrestricted access to food and water post-surgery. Euthanasia was performed if mice appeared moribund or in discomfort, etc., according to the study’s criteria. All experimental data were included in the analysis with the following exceptions: (1) experimental failure, when the blinded operator notes a technical failure in the anesthesia, surgery, blood sampling, therapy administration, etc.; (2) unsuccessful middle artery occlusion as noted by a failure of the hemispheric blood flow to decline by at least 80% or if pathologic analysis shows that the 7–0 monofilament was not correctly placed in the proximal MCA; (3) if there was a loss of sample for measurement or analyses. Overall, a total of 33 successful surgeries out of 36 mice were observed.

### 4.3. Assessment of Neurologic Motor Skills

#### 4.3.1. Neurologic Score

In these set of experiments, mice were subjected to 1–4 h of transient MCA occlusion and were evaluated for the neurologic deficit at 24 h via a 3-point scale, and corner tests were employed [50,51]. The scores were recorded as follows: 0, no deficit; 1, failure to extend the left forepaw fully; 2, circling to the left side; 3, no spontaneous locomotor activity, with a depressed level of consciousness.

#### 4.3.2. Corner Test

A mouse was placed in their own cage between two 30 × 20 × 1 cm boards positioned at a 30° angle to each other, with a small corner opening between them. When the mouse enters the corner, both vibrissae are stimulated together, which provokes the mouse to rear and turn back to face the open end. The number of turns toward each side are recorded in 10 rearing events. Normal mice turn both left and right, but mice with ischemic brain injury tend to turn toward the side of brain injury due to weakness on the contralateral side.

### 4.4. Analyses of Brain Hemorrhage, Infarction, and Swelling

Immediately after the euthanasia and perfusion, the brains were isolated and sectioned coronally with a brain slicer into 2 mm sections in a rostral-caudal orientation. Both faces of the brain slices were immediately digitally photographed through a microscope. Brain slices were promptly incubated in 2, 3, 5-triphenyl tetrazolium chloride (TTC, 2%) to assess cellular viability, followed by digital photography as above. Digital microscopic images were analyzed by a blinded observer using Image Pro Plus 6.2 software to measure areas of brain hemorrhage, TTC staining, and hemisphere swelling. To determine the percentage hemisphere infarction, the TTC-stained areas of the ischemic and non-ischemic hemispheres were measured on both faces of each brain slice. The percentage infarction was calculated for each brain by the formula: infarct percentage = 100 × (V_C_ − V_L_/V_C_), where V_C_ = TTC-stained area in the control hemisphere × slice thickness, V_L_ = TTC-stained area in the infarct hemisphere × slice thickness [52]. Consistent with expert recommendations, the percentage brain hemorrhage in the infarct hemisphere was determined by measuring the area of hemorrhage in digital microscopic images on both sides of each brain slice for the ischemic and contralateral, unaffected control hemisphere (in which there was no hemorrhage) [53]. The percentage hemorrhage = 100 × (volume of hemorrhage in the infarcted hemisphere/volume of the control hemisphere). The amount of swelling in the ischemic hemisphere was determined by comparing the volume of the ischemic hemisphere and the contralateral hemisphere for both faces of each brain slice. The percentage swelling was determined for each brain by the formula: swelling percentage = 100 × (volume of the infarcted hemisphere − volume of the control hemisphere)/volume of the control hemisphere.

### 4.5. Tissue Histology, Immunofluorescence Staining, and Image Analysis

Brain tissues were fixed in 4% paraformaldehyde for 24 h, and paraffin-embedded brain sections (5 μm) were used for immunofluorescence staining as described (45, 50). Sections were deparaffinized (Safe Clear II, Fisher Diagnostics, Kalamazoo, MI, USA), and antigen retrieval was performed by heat-induced epitope retrieval at 98 °C in 10 mM sodium citrate buffer, pH 6.0 for 20 min. After blocking with 10% normal donkey serum for 1 h at room temperature, the sections were incubated with the primary antibody in 2% normal donkey serum overnight at 4 °C followed by 45 min incubation with fluorophore-conjugated donkey secondary antibodies. The primary antibody was goat anti-mouse MMP-9 (#AF 909, R&D systems, Minneapolis, MN, USA), goat anti-mouse MMP-2 (#AF 1488, R&D systems, Minneapolis, MN, USA), goat anti-mouse MMP-3 (#AF 548, R&D systems, Minneapolis, MN, USA), rabbit anti-mouse Collagen IV (#2150–1470, Biorad, Hercules, CA, USA), rat anti-mouse Ly6G (#127602, Bioligend, San Diego, CA, USA), and rat anti-mouse CD68 (#MCA1957, AbD serotec, Hercules, CA, USA); and the secondary antibody was Alexa Fluor^®^ 555 donkey anti-goat antibody and Alexa Fluor^®^ 488 donkey anti-rabbit or -rat antibody. The slides were mounted with Vectashield hard-set mounting media (#H-1500, Vector Laboratories, Burlingame, CA, USA) containing DAPI for nuclei staining.

After staining, all sections were observed and scanned by a blinded experimenter using a Keyence BZ-X800E Microscope (Keyence, Itasca, IL, USA). To analyze the expression levels of MMPs, the entire stroke hemisphere and non-stroke hemisphere were scanned at 10× magnification using consistent excitation/exposure settings. To specifically evaluate the cellular source of MMPs in the mouse brain, double immunostaining using specific antibodies mentioned above and representative fields at 20× or 40× magnification were taken and used for analysis. Quantification of the positively stained area in the hemisphere (stroke vs. contralateral hemisphere) was performed with image analysis software (Image-Pro Plus, version 6.2, Media Cybernetics, Rockville, MD, USA) with the images set to a constant color range in the histogram mode analysis.

### 4.6. Statistics

Data were expressed as means ± SEM. Data were analyzed by an unpaired *t*-test and one-way or multi-way ANOVA with corrections for multiple comparisons. To evaluate the effects of ischemia duration, brain hemisphere (ischemic vs. non-ischemic), and treatment (control vs. tPA) on MMPs (MMP-9, MMP-3, and MMP-2) expression, we performed a three-way ANOVA using ischemic time, hemisphere, and treatment as the independent variables. Additionally, interactions between each pair of factors (hemisphere and ischemic time; treatment and ischemic time; hemisphere and treatment) were assessed to provide a more comprehensive analysis. In some cases, data normality was assessed by Anderson–Darling test and subjected to cos (Y) transformation to achieve normality. The analysis was performed using GraphPad Prism software 8 (La Jolla, CA, USA). Significance was determined at *p* < 0.05. Depending on the parametric nature of the data, Spearman correlation (*rs*) or Pearson correlation (*rp*) analysis was used to analyze possible association between MMPs expression levels and infarct volume, neurologic deficit score, swelling, and hemorrhage.

## 5. Conclusions

Our findings indicate that in experimental ischemic stroke with reperfusion, the duration of ischemia and r-tPA treatment significantly impacts the expression of MMP-2, MMP-3, and MMP-9, as well as the extent of ischemic brain injury and neurological outcomes.

## Figures and Tables

**Figure 1 ijms-25-09442-f001:**
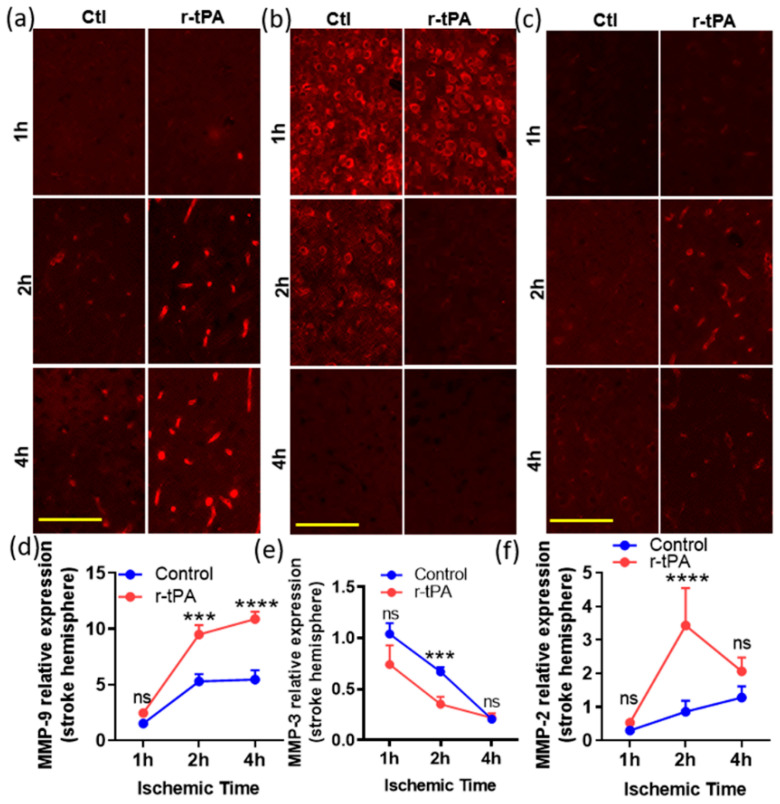
Time-dependent changes in the expression of MMP-9, -3, and -2 in the ischemic (stroke) hemispheres in mouse brains post stroke. The immunofluorescent images at higher magnification (20×, scale bar = 100 μm) in (**a**–**c**) are representative fields showing the expression of MMP-9, 3, and 2 (red) in the ischemic hemisphere at 24 h post stoke with various ischemic times, 1, 2, and 4 h, respectively. The corresponding graphs at the bottom (**d**–**f**) show the quantitation of MMP-9, 3, and 2 expression levels in stroke hemisphere in mice treated with saline (control, ctl) or r-tPA. Data represent means ± SEM. Differences between groups were assessed by one-way ANOVA. *** *p* < 0.001, **** *p* < 0.0001 vs. ctl group at indicated time points. *n* = 4–6. *ns* = non-significant.

**Figure 2 ijms-25-09442-f002:**
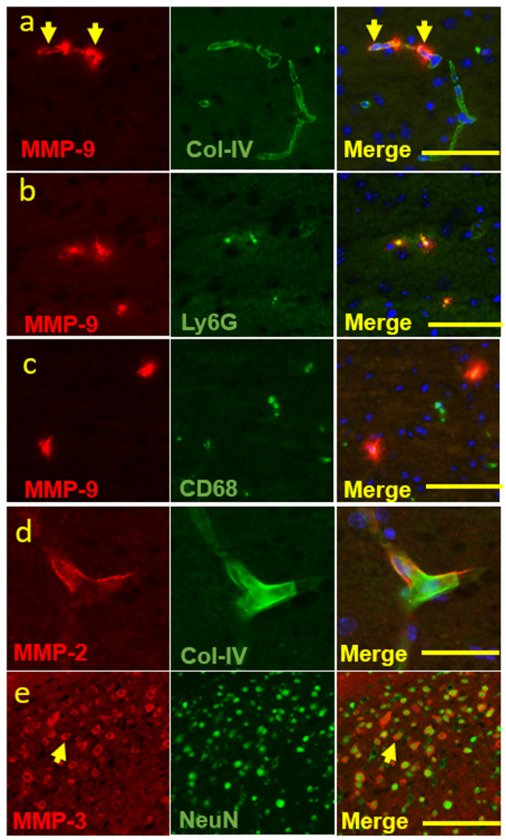
Co-localization of MMP-9, -2, and -3 and various cellular markers. Double immunofluorescence staining experiments were conducted with MMP-9 and various cell markers. (**a**–**c**) MMP-9 and collagen IV or Ly6G or CD68 (originally 20×, scale bar = 50 μm), to show that MMP-9 was expressed in (**a**) the vascular compartments and inflammatory cells, such as (**b**) neutrophil and (**c**) macrophage. (**d**) MMP-2 and collagen IV (originally 40×, scale bar = 25 μm), to show that MMP-2 was expressed in the vascular compartments. (**e**) MMP-3 and NeuN (originally 10×, scale bar = 100 μm), to show a neuronal source of MMP-3 expression. Arrows indicate vascular location of MMP-9 (**a**) and cytoplasmic location of MMP-3 on neurons (**c**) respectively.

**Figure 3 ijms-25-09442-f003:**
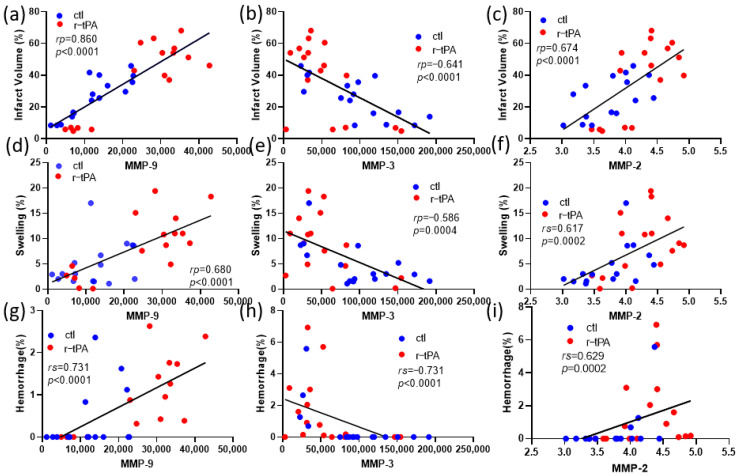
Correlation between expression of MMPs and neuronal outcomes. (**a**,**d**,**g**) Correlation between MMP-9 expression levels in both ctl and r-tPA groups and (**a**) infarct volume, (**d**) swelling, and (**g**) hemorrhage. (**b**,**e**,**h**) Correlation between MMP-3 expression levels in both ctl and r-tPA groups and (**b**) infarct volume, (**e**) swelling, and (**h**) hemorrhage. (**c**,**f**,**i**) Correlation between MMP-2 expression levels in both ctl and r-tPA groups and (**c**) infarct volume, (**f**) swelling, and (**i**) hemorrhage. Data represent means ± SEM of *n* = 12–18 mice per group. Either Pearson or Spearman test was used when the variables were normally distributed or non-parametrically distributed.

**Figure 4 ijms-25-09442-f004:**
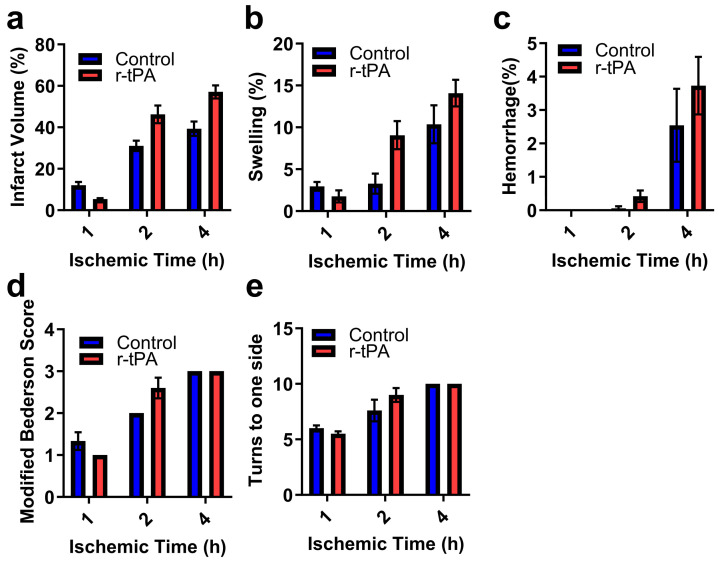
The effect of r-tPA and duration of ischemia on ischemic brain injury and neurobehavioral outcomes. Ischemia was induced by 7–0 monofilament occlusion of the middle cerebral artery for the indicated times, followed by reperfusion for 24 h after ischemia onset. Ischemia was confirmed by an ~80% drop in hemispheric blood by laser Doppler flowmetry. (**a**) The percentage of brain infarction, (**b**) brain swelling (percentage), (**c**) the percentage of brain hemorrhage area. Neurobehavioral disability and sensorimotor dysfunction were assessed at 24 h by (**d**) modified Bederson score and (**e**) Corner test. Data are represented as mean ± SEM. Differences were assessed by two-way ANOVA with the Holm–Sidak correction for multiple inferences. Ctl (*n* = 4–6) and r-tPA (*n* = 5–6) per group.

**Table 1 ijms-25-09442-t001:** Three-way ANOVA analysis of various factors’ effect on MMPs expression.

	Individual Factors					
	Ischemic Time (A)		Treatment, Ctl vs. r-tPA (C)		Hemisphere, NSH vs. SH (B)	
	*p* Value	%TV *	*p* Value	%TV	*p* Value	%TV
MMP-9	<0.0001	24.7	<0.0001	11.1	<0.0001	25.2
MMP-2	0.0003	15.1	0.0016	8.7	0.007	6.2
MMP-3	<0.0001	36.2	0.0153	4.7	0.0003	11.1

* %TV—Percent of total variation.

## Data Availability

The raw data supporting the conclusions of this article will be made available by the authors on request.

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
