# Peer review of "Differences in Acute Expression of Matrix Metalloproteinases-9, 3, and 2 Related to the Duration of Brain Ischemia and Tissue Plasminogen Activator Treatment in Experimental Stroke"

_ijms, 2024, doi:10.3390/ijms25179442_

Round 1

Reviewer 1 Report

Comments and Suggestions for Authors

GENERAL COMMENTS

This is an interesting and clinically relevant experimental study on the role of MMP in the treatment of stroke. However, the points below need to be addressed, in particular to enhance the clinical perspectives of the study, and clarify some formal and methodologic issues.

ABSTRACT/TITLE

The title and abstract are overall appropriate, thank you.

Please mention limitations in the abstract

INTRODUCTION

-       Is there any preliminary or indirect evidence in human patients on the topic?

METHODS

-       It is not clear for which confounding factor each analysis is adjusted, please include more details on the adjusting in each analysis.

-       Were all results corrected for multiple comparisons? If not, specify for each result if it was or was not corrected

-         The authors should describe how outliers were handled, if any.

-       The handling of missing data should be addressed, if any.

RESULTS

The results are overall well-presented, thank you.

DISCUSSION/CONCLUSION

-       The authors should update the limitations according to the points above and the fact that no result can be drawn on females.

-       I regret to say that discussion of clinical and research perspectives is missing. To enhance the findings, please discuss more in depth clinical perspectives and avenues of future research. For instance, MMP emerge as a potential targets for future add-on stroke therapies, however this is not limited to r-tPA, but could be relevant to be studied also in other therapeutical approaches, such as remote ischemic conditioning for instance, in stroke but also myocardial infarction that shares many similarities with stroke (e.g. refer to Remote Ischemic Conditioning in Ischemic Stroke and Myocardial Infarction: Similarities and Differences, DOI: 10.3389/fneur.2021.716316). This, or other avenues of future clinical or experimental research should be included to highlight the relevance of the authors’ findings.

Comments on the Quality of English Language

MINOR

Please correct typos throughout the manuscript

Author Response

We would like to express our sincere thanks to the reviewers for their constructive and insightful comments and suggestions regarding our work. We have carefully addressed all the comments and made improvements to enhance the quality of the manuscript. For a detailed response, please refer to the attached file.

Reviewer 2 Report

Comments and Suggestions for Authors

Matrix Metalloproteinases (MMPs) are a family of zinc-dependent proteolytic enzymes that degrade various components of the extracellular matrix (ECM) and play crucial roles in physiological processes such as tissue remodeling, inflammation, and cell migration. In the context of brain ischemia and tissue plasminogen activator (tPA) treatment, MMPs have significant implications for both injury and recovery. This manuscript studied the expression of MMP-9, 3 and 2 and its colocalization in the ischemic hemispheres in mouse brains post stroke and r-tPA treatment, as well as the correlation between expression of MMPs and neuronal outcomes. These data increased our knowledge about the time-dependent effects of duration of ischemia and r-tPA treatment on brain injury, neurobehavioral outcomes, and the expression of MMPs in cerebral ischemia with reperfusion. However, there are several major and minor weaknesses in the rationale and research methods of this work. Below please find the review comments. 

(1) Major comments

1. The major concern about this manuscript is the insufficient evidence to support the results in the main text. Such as the result 1, to address MMPs expression in the brain after ischemia and r-tPA treatment, it is crucial to supply the super robust IHC images with a clear signal of each protein (antibody), it is also needs to confirm the specificity of the MMP antibody in the IHC imaging, as there is no specific marker to label the cell types or nuclear, it is impossible to quantify the colocalization and quantify the expression levels in different region or cell types. Meanwhile, a whole brain view (IHC image) is required to confirm the conclusion, otherwise, the confocal images in figure 1 are merely limited to some specific spots rather than global hemispheres. And the other techniques measuring the protein and mRNA levels are also required to confirm the expression changes.

2, Based on my knowledge, MMPs play a critical role in the pathophysiology of brain ischemia by contributing to BBB disruption, ECM degradation, and cell death. While tPA remains the standard treatment for acute ischemic stroke, its interaction with MMPs poses significant risks. It is an interesting topic to describe the comparative expression levels of MMP-9, -3 and -2 in the brain post stroke, while it seemed not suitable for the clinical application without any interference or manipulation experiments.

3. It appears that the author tries to examine the correlation between expression of MMPs and neuronal outcomes with N=12-18 mice data per group in figure 3, however I do not think the number is sufficient for the Pearson or Spearman test.

Comments on the Quality of English Language

NA

Author Response

(The authors gave the same response as above.)

Round 2

Reviewer 2 Report

Comments and Suggestions for Authors

appreciate the author's responses to my queries and the additional experimental work they have done to address these. I agree to accept the current manuscript.